# Menstrual Cycle, Glucose Control and Insulin Sensitivity in Type 1 Diabetes: A Systematic Review

**DOI:** 10.3390/jpm13020374

**Published:** 2023-02-20

**Authors:** Elena Gamarra, Pierpaolo Trimboli

**Affiliations:** 1Servizio di Endocrinologia e Diabetologia, Ente Ospedaliero Cantonale (EOC), 6501 Bellinzona, Switzerland; 2Facoltà di Scienze Biomediche, Università della Svizzera Italiana (USI), 6900 Lugano, Switzerland

**Keywords:** menstrual cycle, type 1 diabetes mellitus, continuous glucose monitoring, glucose metrics, insulin sensitivity

## Abstract

The correlation between the menstrual cycle and glucose control in type 1 diabetes has been the focus of several studies since the 1920s, but a few critical aspects made it particularly challenging to reach conclusive evidence. The aim of this systematic review is to reveal more solid information about the impact of the menstrual cycle on glycaemic outcomes and insulin sensitivity in type 1 diabetes and highlight the less researched areas. The literature was searched by two authors independently using PubMed/MEDLINE, Embase and Scopus (last search on 2 November 2022). The retrieved data did not allow us to perform a meta-analysis. We included 14 studies published between 1990 and 2022, with sample sizes from 4 to 124 patients. We found a wide heterogeneity in the definition of the menstrual cycle phases, glucose metrics, techniques for determining insulin sensitivity, hormonal assessment and other interfering factors considered, with an overall high risk of bias. There is no conclusive evidence, and published data do not allow us to achieve quantitative results. In a subset of patients, a possible worsening of insulin sensitivity and hyperglycaemia in the luteal phase could be observed. From the clinical standpoint, a cautious strategy based on patient-specific patterns can be considered until new, solid evidence is obtained.

## 1. Introduction

The relationship between diabetes mellitus, sexual function and the menstrual cycle (MC) came out in the early 1920s, when the suspicion that there might be a further factor potentially impacting glycaemic control in women with type 1 diabetes mellitus (T1DM) was raised [1]. Since then, evaluating this potential correlation has been a challenge due to several critical aspects: (1) the need for intensive glucose monitoring, which has become accurate only in recent decades thanks to the implementation of flash and continuous glucose monitoring systems [2]; (2) a lack of uniformity for estimating insulin sensitivity (IS) [3,4,5,6,7]; (3) the number of variables and determinants to be considered when monitoring patients over a medium-to-long period (i.e., carbohydrate intake, the frequency, duration and intensity of physical activity, premenstrual syndrome (PMS) symptoms and the quality and quantity of sleep) [8,9,10,11]. Currently, the idea that, in a subset of T1DM women, the MC has an impact on glycaemic control consisting of (1) increased exposure to and risk of hyperglycaemia and (2) deterioration of IS in the luteal phase compared to the follicular one has been gaining momentum in clinical practice. However, the literature in this field is discordant, and the pathophysiological mechanism underlying IS changes observed during different phases of the MC remains still unclear [12,13,14]. In fact, some authors found that IS might hold a role, but these data were not confirmed (or not evaluated) by other studies [7,15,16]. Additionally, a wide spectrum of intra- and interindividual variables have to be considered. To summarize, we face in clinical practice T1DM women requiring therapy adjustment during different phases of the MC because of incomplete glycaemic control, but we have no solid information about this phenomenon.

Following the above critical issues, this systematic review aimed to reveal the most robust evidence regarding the impact of the MC on glycaemic outcomes and IS in T1DM women and highlight the less researched areas worthy of further investigation through new targeted studies.

## 2. Materials and Methods

### 2.1. Conduct of Review

The systematic review was performed in accordance with Preferred Reporting Items for Systematic Reviews and Meta-Analyses (PRISMA) statement [17].

### 2.2. Search Strategy and Study Selection

We searched for studies reporting direct/indirect measurement of IS and/or glycaemic control parameters in women with T1DM along the different phases of the MC. A comprehensive computer literature search of the PubMed/MEDLINE, Embase and Scopus databases was conducted to find published articles on this topic. The search algorithm was based on the combinations of keywords and MeSH terms: ((type 1 diabetes) oe (IDDM) OR (insulin-dependent diabetes) OR (type 1 diabetic women)) AND (menstrual cycle) AND ((insulin sensitivity) OR (insulin requirement) OR (insulin delivery) OR (insulin dosing) OR (insulin secretion) OR (glucose effectiveness) OR (glucose infusion rate) OR (glucose test) OR (clamp)) AND ((glycaemic outcomes) OR (metabolic outcomes) OR (glucose control) OR (metabolic control) OR (glucose metabolism) OR (glucose variability) OR (glycaemic variability) OR (hyperglycaemia) OR (metabolic profile)). We did not use a beginning date limit, and the search was updated until 2 November 2022. To identify additional studies and expand our search, the references of included articles were also screened. Articles known by the authors could be included even if they were not found by the database search. All original articles that described variations in glucose control and/or IS during the MC in T1DM women were eligible for inclusion. Case reports were excluded. No language restriction was adopted. Two investigators (EG and PT) independently searched for papers, screened the titles and abstracts of the retrieved articles, reviewed the full texts and selected articles for inclusion. Discordances were solved in a mutual consensus.

### 2.3. Data Extraction

The following information of included studies was independently extracted by the same two investigators in a piloted form: (1) general information such as the author(s), year of publication, aim, study type, number of patients and number of MCs, inclusion and exclusion criteria and definition and determination of MC phases; (2) outcome measures and methods, including glucose metrics, direct/indirect IS metrics, hormonal and metabolic blood tests and timing of blood samples, food intake (carbohydrate and caloric intake), physical activity, sleep quality and premenstrual syndrome symptoms; and (3) results.

### 2.4. Study Quality Assessment

The risk of bias in included studies was independently assessed by the two authors according to the National Heart, Lung and Blood Institute Quality Assessment Tool for Observational Studies [18].

## 3. Results

### 3.1. Eligible Articles

After excluding duplicates, the online search retrieved 181 articles. According to the above selection criteria, 19 articles were initially selected. Of those, 7 were excluded for an inadequate population studied, overlapping data, lacking data or study design, and 12 were included. Another two studies were also included, one selected from the references of retrieved articles [19] and one known by the authors of this review [9]. Finally, 14 studies [7,10,11,12,13,14,15,16,20,21,22,23] were included in the systematic review (Figure 1).

### 3.2. Qualitative Analysis

The main features of the 14 studies are summarized in Table 1. The 14 articles were published between 1990 and 2022 in scientific journals in the fields of diabetology-endocrinology and nursing care. Overall, the sample size ranged from 4 to 124 patients. Table 2 illustrates the methodological framework and the major results, and more details about the glucose metric, IS metric, laboratory tests and other evaluations performed in the studies are available in Appendix A (Table A1). The number of MCs observed ranged from 1 to 6.5/patient (total number from 6 to 168 MCs). The definition of the different phases of the MC among studies was heterogeneous, i.e., a simple division into the follicular and luteal phase or the identification of subphases (early follicular, mid-late follicular, periovulation, early luteal, mid-luteal and late luteal). In addition, the determination of phases was sometimes (eight studies) simply based on dates of menses reported by women and calculated considering the average duration of an MC (28 days) with the ovulation occurring exactly on day 14. In other cases (five studies, not necessarily the most recent ones), it was established through blood/urine hormonal tests in order to identify ovulation and the fluctuations of sex hormones that characterize the different phases (estradiol, progesterone, LH and FSH).

Studies were conducted using a wide range of glycaemic metrics for the assessment of metabolic outcomes and heterogeneous methods for determining IS. To define women’s glucose control throughout the MC, most recent papers (since 2004) used continuous glucose monitoring parameters such as mean glucose, time in/above/below range (TIR/TAR/TBR, respectively), the coefficient of variation (CV), the Kowatchev high/low blood glucose index (HBGI and LBGI, respectively) and average daily risk range (ADRR). Previous articles based this evaluation on patient self-reported glucose values or capillary tests, from which, in some cases, mean glycaemia and standard deviation (SD) were obtained. Only four studies used the definition of hypo- and hyperglycaemia according to the most recent international recommendations. A total of 11 studies have evaluated IS using different direct or indirect metrics: in 4 studies, the total daily dose (TDD) was evaluated, in most cases coupled with carbohydrate/food intake, whereas 7 studies used direct methods such as the euglycaemic clamp, the frequently sampled intravenous glucose tolerance test (FSIGT), the insulin-glucose infusion test (IGIT), the hyperglycaemic hyperinsulinemic clamp or the Kalmann filtering method from closed-loop meal and insulin data.

As an additional factor of heterogeneity, three studies also measured the levels of some hormones potentially influencing the difference in IS found across the phases of the menstrual cycle, such as cortisol, norepinephrine, GH, testosterone, DHT and androstenedione.

Finally, in five articles, other factors are considered and other data are collected through wearable trackers, questionnaires or validated scales (Penn Daily Symptoms Rating Scale and Pittsburg Sleep Quality Index): body temperature, weight, occasional medications, physical exercise, sleep quantity and quality, emotional state, psychosocial condition and premenstrual symptoms.

### 3.3. Quality Assessment

The following aspects were evaluated: study questions; eligibility criteria; sample size, assessment of exposure (MC) and outcomes; and statistical methods. Overall, the risk of bias was high, especially concerning the sample size, analysis of the power of the sample, timeframe, exposure definition and assessment and potential confounding variables. These results are detailed in Table 3.

Questions:

1. Was the research question or objective in this paper clearly stated?

2. Was the study population clearly specified and defined?

3. Was the participation rate of eligible persons at least 50%?

4. Were all the subjects selected or recruited from the same or similar populations (including the same time period)? Were inclusion and exclusion criteria for being in the study prespecified and applied uniformly to all participants?

5. Was a sample size justification, power description or variance and effect estimates provided?

6. For the analyses in this paper, were the exposure(s) of interest measured prior to the outcome(s) being measured?

7. Was the timeframe sufficient so that one could reasonably expect to see an association between exposure and outcome if it existed?

8. For exposures that can vary in amount or level, did the study examine different levels of the exposure as related to the outcome (e.g., categories of exposure, or exposure measured as a continuous variable)?

9. Were the exposure measures (independent variables) clearly defined, valid, reliable and implemented consistently across all study participants?

10. Was the exposure(s) assessed more than once over time?

11. Were the outcome measures (dependent variables) clearly defined, valid, reliable and implemented consistently across all study participants?

12. Were the outcome assessors blinded to the exposure status of participants?

13. Was loss to follow-up after baseline 20% or less?

14. Were key potential confounding variables measured and adjusted statistically for their impact on the relationship between exposure(s) and outcome(s)?

## 4. Discussion

Increasing scientific efforts try to focus on and enlighten the correlations between menstrual cycles and glycaemic control and the impact of the former on the latter in women with T1DM. Just as some (but not all) T1DM individuals experience a “dawn phenomenon” that requires insulin adjustment, a subset of T1DM women (but not all of them) experience a “menstrual cycle phenomenon” [23], which can represent an obstacle to the achievement of optimal glycaemic control and negatively impact patients’ quality of life. This study was conceived to better inform decision making regarding glycaemic management and help patients cope with glucose control problems by modifying something in their daily lives, if the situation so demands.

Based on the published literature, there is no conclusive evidence on this topic. However, as summarized in Table 4, some questions can be addressed. Regarding glycaemic control, (1) nine studies show worsening or higher risk of hyperglycaemia in the luteal phase versus the follicular phase, expressed in terms of TAR [9—SS, 10—SS, 14—NS, 19—NS, 22—SS, 23—NS], mean SMBG glycaemia/CGM glucose [9—SS, 19—SS], glycaemic response to FSIGT [23—NS], HBGI [12—SS], 2h postprandial glycaemic peak [22—SS] and glycaemic response to the hyperinsulinemic hyperglycaemic clamp [15—NS], and (2) two studies [11,21] show unchanged glucose metrics across phases of the MC, checked by CGM or SMBG. Regarding IS, (1) four studies show worsening of IS in the luteal phase compared to the follicular phase, estimated by the euglycaemic clamp [20—SS], Kalman filtering method [12—SS], FSIGT [23—NS] and hyperglycaemic hyperinsulinemic clamp [15—NS], and (2) two studies [7,8,9,10,11,12,13,14,15,16] show the absence of SS differences in IS in the various phases of the MC, estimated by IGIT or the euglycaemic clamp.

Unfortunately, the retrieved data did not allow us to perform a meta-analysis to achieve quantitative results. In addition, methodological critical issues were found in the included studies. First, sample sizes were generally small. Second, the methods to define both MC phases and their duration were heterogeneous. Third, glycaemic metrics were also heterogeneous, with only a few studies using CGM metrics as recommended by international guidelines [24] and often over a limited period of observation. Fourth, the IS evaluation methods were questionable. The evaluation was carried out with euglycaemic or hyperglycaemic clamp methods, with the potential to influence the result. Previous research has indeed used various techniques to assess IS during the MC such as the glucose tolerance test and the euglycaemic or hyperglycaemic clamp [25,26,27,28,29,30,31,32], but most of the studies have been performed in nondiabetic women and have obtained different results with the hyperglycemic versus the euglycemic technique. Therefore, applying their results to T1DM women requires much caution, as the euglycaemic range of nondiabetic subjects does not reflect the real relatively hyperglycaemic state that characterizes T1DM patients and can impact glucose tolerance [33]. Furthermore, IS refers to daytime, night-time or both in different studies.

According to the retrieved data in the systematic review, further issues should be addressed.

First, the prevalence of the “menstrual cycle phenomenon” among women with T1DM is unclear, and data from individual studies are not generalizable to a large population of T1DM patients.

Second, the phenotype of the glycaemic worsening found in the luteal phase is unclear. Since, according to some studies, this worsening affects postprandial glucose [13,19,22], it would be preferable to change meal boluses and the IS factor for corrections rather than basal insulin. However, according to other studies, the worsening occurs mainly in fasting glucose due to the accentuation of the dawn phenomenon caused by progesterone [22,34], with adjustment of basal insulin being the preferred strategy. The effect of oestrogen and progesterone on IS and glycaemic parameters remains unresolved and could, at least partially, explain blood glucose/IS fluctuations. On the one hand, estradiol may contribute to the increased risk of hyperglycaemia, and its lowering increases the risk of hypoglycaemia [15]. On the other hand, the increase in progesterone in the luteal phase would determine the increased risk of hyperglycaemia [13,23]. In addition, the potential impact of progesterone on caloric/CHO intake is still a matter to debate, and whether a fixed combination of OCPs can help smooth out glucose fluctuations remains to be investigated [13]. Finally, it is still unclear whether fluctuations of other hormones (such as GH, steroids and androgens) over the MC are related to the variability in glucose control [13,16]. The debated mechanisms through which hormones contribute to IS alterations involve the binding affinity of the insulin receptor and/or some other postbinding defect in the various phases of the MC [15,35,36,37].

Third, the potential role of glucose control itself in the MC has been described in T1DM adolescents, in whom higher HbA1c is associated with greater cycle irregularities [38]. On the other hand, the availability of new technologies has increased the possibility to optimize glycaemic control through the tailoring of insulin treatment, resulting in a potential attenuation of menstrual-associated variability.

Fourth, other factors, such as lifestyle parameters and diabetes care habits (i.e., physical activity and exercise, food intake and sleep quality and duration), have to be taken into account.

With these premises, we advise future studies to have these features: a prospective, preferably randomized, design, with adequate sample size calculation, including patients in OCPs, standardized glucose monitoring and IS estimation metrics, observed changes in glycaemic control in fasting and postprandial conditions, including data on insulin administration (AHCL, CSII and MDI), and the collection of data on lifestyle and diabetes care habits. In addition, data on sexual hormones, GH, steroids and androgens should be collected.

## 5. Conclusions

Both questions raised by the present study remain partially unsolved, and published data do not allow us to achieve quantitative results. From a clinical standpoint, insulin therapy has to be tailored in women experiencing premenstrual hyperglycaemia. Until new, solid evidence is obtained on this topic, patient-specific patterns should be assessed by continuous glucose monitoring. Then, different parameters (i.e., the basal rate, insulin/carbs ratio and correction factor) have to be adapted according to data recorded on different phases of the MC in a similar manner to presetting for working vs. nonworking days. Alternatively, real-time self-management based on glucose changes may be considered in highly compliant patients, particularly in order to reduce the risk linked to the intrapersonal variability of the MC, potentially making any presetting not always suitable.

## Figures and Tables

**Figure 1 jpm-13-00374-f001:**
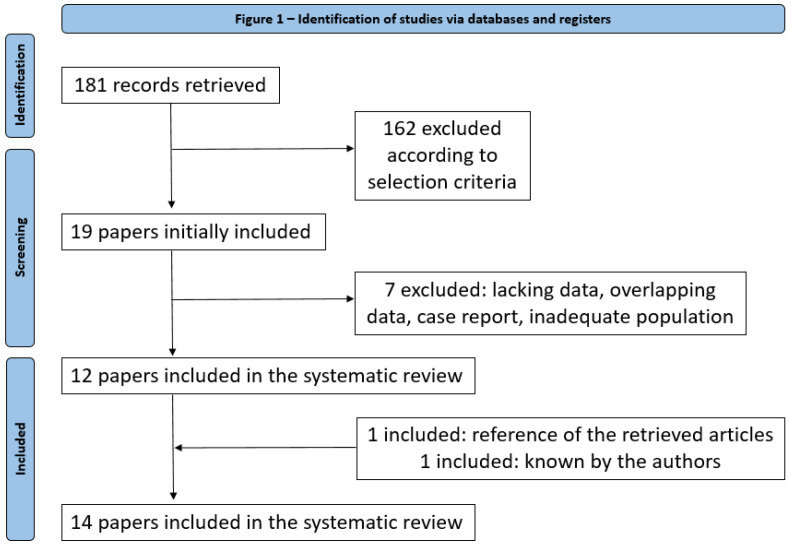
Diagram of flow of articles.

**Table 1 jpm-13-00374-t001:** Study characteristics.

Authors	Year of Publication	Journal	Sample Size				Inclusion Criteria	Exclusion Criteria	Aim
			Nr Patients Enrolled	Nr Patients in OCP	Nr MC/pat	Nr MC tot			
Tatulashvili et al. [10]	2022	*J. Clin. Endocrinol. Metab.*	24	0	1–3	62	T1D, > 18 years, premenopausal, spontaneous MC of 24–35 days, use of CGM, MDI/CSII	Lifestyles/therapy changes in the 3 months prestudy, HbA1c > 10%, OCP, pregnancy, PCOS, gastroparesis, language barriers	Evaluate the variability in glucose values according to different phases of menstruation through CGM data in T1D women.
Diaz et al. [20]	2022	*Diabetes Technol. Ther.*	16	8	1	16	T1D since > 1 year, 12–35 years, HbA1c < 9%, regular MC	Pregnancy or breastfeeding, any OCP if 12–17 years, only P OCP if 18–35 years	Test the hypothesis that improved glycaemic control can be achieved in women experiencing premenstrual hyperglycaemia if a priori knowledge on cycle-related IS changes is properly considered when planning insulin therapy
Levy et al. [21]	2022	*Diabetes Technol. Ther.*	16	4	3–14	96	T1D since > 1 year, > 14 years, TDD > 10 UI/die, use of Tandem C-IQ	Pregnancy, other diabetes medications	Analyse insulin delivery and glycaemic metric throughout the menstrual cycle for women with T1D using closed-loop control insulin delivery
Herranz et al. [19]	2016	*Med. Clin. (Barc.)*	26	0	6.5 ± 2.2	168	T1D, CSII plus SMBG	OCP	Determine the frequency of women with T1D showing menstrual cycle changes in glycaemia, analyse their clinical characteristics and assess the pattern of glycaemic changes
Gamarra et al. [9]	2016	*J. AMD*	10	1	4	40	T1D since > 5 years, 17–40 years, CSII plus CGM since > 1 year, HbA1c 6–8.5%, eumenorrhea since > 1 year or OCP, BMI < 30 kg/m^2^	Diabetes complications (except nonproliferative diabetic retinopathy), pregnancy or breastfeeding	Point out correlations between female menstrual cycle, sleep quality and glycaemic response in a sample of T1D patients in order to find which ones deserve further investigation to become clinically relevant
Brown et al. [12]	2015	*J. Diabetes Sci. Technol.*	12	0	3	36	T1D since > 2 years, CSII since > 6 months, HbA1c 5–10%, regular MC 20–40 days	Pregnancy/desire for, OCP/intrauterine device/Depo-MAP, PCOS, elevated testosterone, paracetamol use, steroid use for > 10 days, uncontrolled thyroid disease, elevated liver enzymes	Identify glycaemic variability and IS changes using data available in the outpatient setting (of T1D) across menstrual cycle phases
Barata et al. [22]	2013	*Diab. Care*	6	0	1	6	T1D, regular MC, HbA1c < 8%, MDI/CSII	OCP, pregnancy or breastfeeding, impaired thyroid function	Evaluate the effect of menstrual cycle in T1D patients
Trout et al. [23]	2007	*Diabetes Technol. Ther.*	6	0	1	6	T1D, 18–45 years, HbA1c < 9%	OCP or antidepressants in the 2 months prestudy, significant comorbidities, significant life changes, pregnancy or breastfeeding	Evaluate possible differences in IS between follicular and luteal phases in women with T1D
Goldner et al. [13]	2004	*Diabetes Technol. Ther.*	4	0	3	12	T1D, 18–45 years, HbA1c < 7.5%, creatinine < 1.5 mg/dl, normal proteinuria, MDI/CSII	OCP, pregnancy or breastfeeding, proliferative diabetic retinopathy, cardiovascular disease, gastroparesis	Describe the pattern of changes in glucose control throughout the complete menstrual cycle and the reproducibility of these changes
Lunt et al. [14]	1996	*Diabet Med.*	124	9	NR	NR	T1D since > 6 months, 18–40 years, OCP/not OCP	Pregnancy	Describe the prevalence and temporal pattern of self-reported changes in capillary glucose and insulin dose during the menstrual cycle (in T1D) and compare HbA1c result between subjects who adjust their insulin dose perimenstrually and those who do not
Moberg et al. [7]	1995	*Diabet Med.*	15	0	1	15	T1D, MDI/CSII	Diabetic retinopathy, nephropathy and peripheral neuropathy	Compare day-to-day variation in IS in males and females with T1D and assess IS in the follicular and luteal phase of the menstrual cycle
Lundmann et al. [11]	1994	*Int. J. Nurs. Stud.*	20	0	1–2	18	T1D since > 2 years, OCP/not OCP	Pregnancy, dialysis	Elucidate the impact of menstruation on metabolic control and daily living in patient T1D
Widom et al. [15]	1992	*Diab. Care*	16	0	1	16	T1D since > 2 years, 18–38 years, HbA1c > 7.5%	OCP	Examine the hormonal mechanisms underlying the variability in glycaemic control during the different phases of the menstrual cycle in women with T1D
Scott et al. [16]	1990	*Diabet Med.*	9	0	1	9	T1D	OCP	Compare IS and cardiovascular function before and during hyperinsulinaemia in young T1D women during the follicular and luteal phases of the menstrual cycle

Table legend: OCP: oral contraceptive, MC: menstrual cycle, T1D: type 1 diabetes, MDI: multiple daily injections, CSII: continuous subcutaneous insulin infusion, CGM: continuous glucose monitoring, IS: insulin sensitivity, TDD: total daily dose, SMBG: self-monitoring of blood glucose.

**Table 2 jpm-13-00374-t002:** Methodological framework and major results—* see Appendix A Table A1 for more details.

Study	MC Phases Definition	MC Phases Assessment	Glycaemic Metrics *	Definition of Hypo/Hyper Glycaemia According to International Guidelines	IS Metric *	Hormonal or Metabolic Lab Tests *	Other Factors Considered *			Outcomes
							Diet	Questionnaires.	Wearable Trackers	
Tatulashvili et al. [10]	EF, MF, PO, ML, LL	Dates of menses	CGM	Yes	Patient reported data	No	Yes	Yes	No	TIR decreases over the phases with SS difference between EF and LL. TAR is SS higher in LL vs. EF. TBR is SS higher in MF vs. EF
Diaz et al. [20]	F and L	NR	Clamp	NA	Direct parameter	No	No	No	No	AUC GIR SS decreases from F to L (equals decrease in IS in L-phase)
Levy et al. [21]	Menstruation, L, all the rest	Dates of menses	CGM	Yes	Indirect parameter	No	No	No	No	CGM and insulin metrics unchanged across cycle phases
Herranz et al. [19]	EF, LF, EL, LL	Dates of menses	SMBG	NA	Indirect parameter	No	No	No	No	65.4% of women had cycle changes, defined as > 0.8 mmol/L (15 mg/dL) increase in mean blood glucose from EF to LL in > 2/3 of cycles, with mean glucose and %SMBG > 7.8 mmol/L (140 mg/dl) SS increase from EF to LL. No SS changes in other parameters
Gamarra et al. [9]	EF, MLF, PO, EL, LL	Dates of menses and urinary LH, P test	CGM	Yes	Indirect parameter	Yes	No	Yes	Yes	SS increase in mean glycaemia and SD in L-PO, higher TBR in F and TAR in EL. Not SS changes in insulin dose, CHO intake and sleep efficiency.
Brown et al. [12]	EF, MLF, PO, EL, ML, LL	Dates of menses and urinary LH, P test	CGM	NA	Direct and Indirect parameter	Yes	Yes	No	No	SS higher HBGI in PO and EL (increases progressively up to EL and then falls back). LBGI stable in F and then decreases (but not SS). IS (nocturnal) SS decreases in L vs. EF. No changes in TDD, CHO and Kcal.
Barata et al. [22]	72 h F (day 4–8) and 72 h L (day 18–22)	US and E2 and P test	CGM	Yes	NA	Yes	No	No	No	TAR increases and the TBR decreases in L vs. F. Mean glucose SS higher 2h post breakfast and 2h post lunch in L vs. F
Trout et al. [23]	F (day 6–8 post menstruation) and L (day 7–9 post urinary test +)	Dates of menses and urinary LH, P and E2 test	Clamp	NA	Direct parameter	Yes	No	No	No	Mean glucose higher in L vs. F (not SS). SI higher in F vs. L (+24%, not SS). IS inversely correlated with P level and Penn Daily scale score, not correlated with E2 and cortisol level
Goldner et al. [13]	L (14 days), O (-14 day from menstruation), F	Dates of menses and P, E2, LH and FSH test	CGM	No	Direct parameter	Yes	Yes	Yes	No	2/4 patients: higher TAR in L. 2/4 patients: no glucose pattern. SS direct correlation between P level and hyperglycaemia in 2/4 patients. SS inverse correlation between E2 level and hyperglycaemia in 1/4 patients. Reproducibility: not between women but between cycles in the same woman
Lunt et al. [14]	Pre/post menstruation.	Dates of menses	Patient reported data	NA	Patient reported data	Yes	Yes	Yes	No	61% of all patients (67% of OCP patients) notice glycaemic changes, especially glucose increase in L. HbA1c does not change between adjusters (36%) vs. nonadjusters (25%) of insulin patients. 56% of patients notice appetite change, especially increased food intake perimenstrually
Moberg et al. [7]	F (days 2–14) and L (days 19–31)	Dates of menses	Clamp	NA	Direct parameter	Yes	No	No	No	Not SS differences in IS L vs. F
Lundmann et al. [11]	F and L	Dates of menses	Patient reported data	NA	Patient reported data	Yes	Yes	Yes	No	Not SS differences in glucose and food intake in L vs. F
Widom et al. [15]	MF (days 5–11) and ML (days 20–29)	Dates of menses	Clamp	NA	Direct parameter	Yes	No	No	No	Not SS worsening of hyperglycaemia and IS in L, associated with E2 level increase only in a subset of patients
Scott et al. [16]	F (days 6–9) and L (days 20–23)	Dates of menses	Clamp	NA	Direct parameter	Yes	No	No	No	Not SS differences in IS in L vs. F, but SS higher GH level in L with possible consequences on glucose production, glucose disposal and IS

Table legend: EF: early follicular, MF: mid-follicular, PO: periovulatory, ML: mid-luteal, LL: late luteal, SS: statistically significant, TIR: time in range, TAR: time above range, TBR: time below range, AUC GIR: under the curve area of the glucose infusion rate, IS: insulin sensitivity, F: follicular, L: luteal, NR: not relevant, NA: not applicable, LF: late follicular, EL: early luteal, SMBG: self-monitoring of blood glucose, SD: standard deviation, CHO: carbohydrates, TDD: total daily dose, OCP: oral contraceptive.

**Table 3 jpm-13-00374-t003:** Risk of bias summary: review of authors’ judgements about each risk of bias item for each included observational study.

First Author, Year	1	2	3	4	5	6	7	8	9	10	11	12	13	14	Total
Tatulashvili, 2022	Yes	Yes	Yes	Yes	No	Yes	Yes	Yes	No	No	Yes	No	Yes	No	9/14
Diaz, 2022	Yes	Yes	Yes	Yes	No	Yes	No	No	NR	No	Yes	No	Yes	No	7/14
Levy, 2022	Yes	Yes	Yes	Yes	No	Yes	Yes	No	No	No	Yes	No	Yes	No	8/14
Herranz, 2016	Yes	Yes	Yes	Yes	No	Yes	Yes	Yes	No	No	Yes	No	Yes	No	9/14
Gamarra, 2016	Yes	Yes	Yes	Yes	No	Yes	Yes	Yes	Yes	Yes	Yes	No	Yes	No	11/14
Brown, 2015	Yes	Yes	Yes	Yes	No	Yes	Yes	Yes	Yes	Yes	Yes	No	Yes	No	11/14
Barata, 2013	Yes	Yes	Yes	Yes	No	Yes	No	No	Yes	Yes	Yes	No	Yes	No	9/14
Trout, 2007	Yes	Yes	Yes	Yes	No	Yes	No	No	Yes	Yes	Yes	No	Yes	No	9/14
Goldner, 2004	Yes	Yes	Yes	Yes	No	Yes	Yes	No	Yes	Yes	Yes	No	No	No	9/14
Lunt, 1996	Yes	Yes	Yes	Yes	No	Yes	NS	No	No	No	Yes	No	Yes	No	7/14
Moberg, 1994	Yes	Yes	Yes	Yes	No	Yes	No	No	No	No	Yes	No	Yes	No	7/14
Lundmann, 1994	Yes	Yes	Yes	Yes	No	Yes	No	No	No	No	Yes	No	No	No	6/14
Widom, 1992	Yes	Yes	Yes	Yes	No	Yes	No	No	No	No	Yes	No	Yes	No	7/14
Scott, 1990	Yes	Yes	Yes	Yes	No	Yes	No	No	No	No	Yes	No	Yes	No	7/14

NR: not relevant, NS: not stated.

**Table 4 jpm-13-00374-t004:** Summary of findings of the present systematic review.

Question of the Present Systematic Review	Conclusion	References Supporting These Findings
What impacts of the MC on glycaemic outcomes can be expected in women with T1DM?	Possible worsening/higher risk of hyperglycaemia in the luteal phase versus the follicular phase in a subset of patients	[9,10,12,13,14,15,19,22,23]
What impacts of the MC on IS can be expected in women with T1DM?	Possible worsening of IS in the luteal phase versus the follicular phase in a subset of patients	[12,15,20,23]

## Data Availability

No new data were created or analysed in this study. Data sharing is not applicable to this article.

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
