# Peer review of "Menstrual Cycle, Glucose Control and Insulin Sensitivity in Type 1 Diabetes: A Systematic Review"

_jpm, 2023, doi:10.3390/jpm13020374_

Round 1

Reviewer 1 Report

Interesting stimulating topic including a sound literature research and analysis with an astonishing result: No clear clinical conclusion can be made from the published literature.

This surely may stimulate new research concerning the impact of menstrual cycle on glucose metabolism in women with Type 1 diabetes mellitus.

Thus, these review should be published.

One typo to be corrected in line 57 of the "Discussion" section: "hypeglycemic" instead of "hyperglycemic" probably.

Author Response

Dear reviewer
Thanks for your review and comments.
The typo in line 57 of the "Discussion" section "hypeglycemic" instead of "hyperglycemic" has been corrected

Reviewer 2 Report

I read with interest this review on glucose control and insulin sensitivity in type 1 diabetes. I found the research topic very interesting.

Research methods are well described and well summarized in the tables.

I am a little disappointed by the conclusions but it is not the fault of the authors of the review. Indeed, the fact that there is no conclusive evidence on this topic is however a very important data as well as a starting point for future research

Unfortunately, the authors cannot provide clear guidance to clinicians on how to behave in women with type 1 diabetes mellitus during the menstrual cycle.Instead, the conclusions provide clear guidance for researchers who would like to explore this topic further.

I only have to ask for a small change:

-  This sentence should be deleted (and the references too)because it is not relevant. The Authors Are Assessing the Exact Opposite Effect of Cycle Hormones On glycemia 

"The potential negative impact of diabetes not well controlled on sexual function and menstruation was already known and some authors reported the impact of MC on carbohydrate tolerance in diabetic women"

Author Response

Dear reviewer
Thanks for your review and comments.
The irrelevant sentence in the introduction has been deleted.